# Interfacial Stability of TiC/γ-Fe in TiC/316L Stainless Steel Composites Prepared by Selective Laser Melting: First Principles and Experiment

**Peikang Bai** [1,*,†], **Qin Wang** [1,†], **Zhanyong Zhao** [1,*], **Wenbo Du** [2], **Minjie Liang** [1], **Haihong Liao** [1], **Yuxin Li** [1], **Lizheng Zhang** [1], **Bing Han** [3] and **Jing Li** [1]

[1] School of Materials Science and Engineering, North University of China, Taiyuan 030051, China; wq950724@163.com (Q.W.); nucliangminjie@sina.com (M.L.); lhh@nuc.edu.cn (H.L.); liyuxin@nuc.edu.cn (Y.L.); 18435841275@163.com (L.Z.); jing.li3d@hotmail.com (J.L.)

[2] National Key Laboratory for Remanufacturing, Army Academy of Armored Forces, Beijing 100072, China; dwbneu@163.com

[3] College of Mechatronics Engineering, North University of China, Taiyuan 030051, China; chihb2008@live.cn

\* Correspondence: baipeikang@nuc.edu.cn (P.B.); zhaozy@nuc.edu.cn (Z.Z.); Tel.: +86-136-2346-6690 (Z.Z.)

† These authors contributed equally to this work.

**Abstract:** TiC/316L stainless steel composites were prepared by selective laser melting. The adhesion work, interface energy and electronic structure of the TiC/γ-Fe interfaces in TiC/316L stainless steel composites were studied to investigate the heterogeneous nucleation potential of γ-Fe grains on TiC particles by using the first principle. The degree mismatch between TiC (001) and γ-Fe (001) interface was lowest. There are four TiC (001)/γ-Fe (001) interface models with different stacking sequences (on-site and bridge-site) and different atomic arrangement sequences (C centre and Ti centre). The results show that the Fe-on-Ti centre interface had the largest work of adhesion (3.87 J/m$^2$) and lower interfacial energy (0.04 J/m$^2$), it was more stable, and the interfacial energy of the model was lower than that of γ-Fe/Fe melt (0.24 J/m$^2$). Strong Fe-C covalent bonds and Fe-Ti metallic bonds were formed near the interface, which increased the interfacial strength, indicating that TiC had strong heterogeneous nucleation potency for γ-Fe.

**Keywords:** laser processing; metals and alloys; first principle; interfacial stability; heterogeneous nucleation

## 1. Introduction

316L stainless steel is widely used in automotive structures, biomedicine, aerospace, machinery, and the chemical and petroleum industries, as well as many other fields, due to its advantages such as good corrosion resistance, strong fracture toughness, high ductility, high strength and biocompatibility [1]. However, ordinary 316L stainless steel materials are not sufficient to resist the requirements of harsh environments, preventing their further development [2]. In addition, hard ceramic particles can be used as the reinforcing phase of iron-based composite materials to improve the mechanical behavior by providing high hardness, high melting point, and high thermal conductivity [3], which effectively expands the application field of 316L stainless steel materials. The most common ceramic reinforcement phases are TiC and TiB$_2$. Sulima et al. [4] studied the reinforcing effect of TiB$_2$ on 316L stainless steel composites with different mass fractions. They found that 1 vol% TiB$_2$ can significantly improve its tensile strength and make its elastic modulus reach 208 Gpa; Zhao et al. and Wang et al. [5] obtained steel-based composites using in situ TiC-TiB$_2$ particles; the main phases were TiC, TiB$_2$ and α-Fe. The results showed that the microhardness and wear resistance performance of the composite was greatly improved with the increase in ceramic

particles. Selective laser melting (SLM), as a promising additive manufacturing technology, can process high-performance metal matrix composites (MMC). In addition, the SLM process has been successfully used to prepare 316L stainless steel [6]. Zhao et al. [7] reported the effect of adding TiC on the microhardness of SLM-formed 316L stainless steel parts. They found that TiC could considerably improve the microhardness and mechanical properties of the 316L stainless steel parts.

The bonding strength between the ceramic particles and matrix determines its mechanical strength and toughness, Zhao et al. [8] introduced the preparation method of graphene-reinforced metal matrix composites, and found that the interfacial bonding strength affected the mechanical properties of the composites. However, it is difficult to measure the composites interface interaction through traditional experiments and reveal the bonding stability between the interfaces from the perspective of atoms [9]. Therefore, theoretical analysis and calculations play an important role in exploring the structural characteristics between interfaces [10]. First principles calculation has been commonly used in recent years. It can perform accurate and effective analysis from the perspective of atoms and electrons [9]. Zhao et al. [11] calculated the interfacial structure of $Al/Al_4C_3$ by using first principles. The results revealed that the C-termination centre interface was the most stable in $Al/Al_4C_3$. Zhang et al. [12] investigated the electronic structures of the TiC (001)/Cu (001) interface using first principles and explored the interfacial bonding strength and stability, finding that the Ti-HS-Cu interface model was the most stable. Zhuo et al. [13] studied the interfacial cohesive energy, interfacial stability and bond properties of Al (111)/$NbB_2$ (0001) based on first principles, and explored the heterogeneous nucleation potential of $\alpha$-Al grains on $NbB_2$. There is a lack of research on interfacial stability between TiC and 316L stainless steel. In addition, many studies have discussed the heterogeneous nucleation potential of Fe on TiC atoms and grain refinement by experiment [1], which is relatively scarce for theoretical analysis.

The $\gamma$-Fe is the main phase in 316L stainless steel, and thus, the adhesion work, interface energy, electronic structure and bonding properties of the TiC (001)/$\gamma$-Fe (001) were explored. The heterogeneous nucleation potential of $\gamma$-Fe on TiC atoms were discussed.

## 2. Computational and Experimental Procedure

A Renishaw AM 250 with a YLR-400 fibre laser (Renishaw, London, UK) was used to prepare the TiC/316L stainless steel composites under Ar gas protection. The parameters following parameters were adopted: the laser power was 200 W, the scanning speed was 1200 mm/s, the laser beam diameter was 70 μm, and the hatch spacing was 60 μm. The microstructure of the composites was characterised with a scanning electron microscope (SEM) (Zeiss Ultra 55, Carl Zeiss Microscopy, Jena, Germany), which was equipped with an energy-dispersive X-ray spectral (EDS) analyser. The accelerating voltage is 0.5–30 kV, and the beam intensity is 1 pA–200 nA.

The calculations were performed using the Cambridge Serial Total Energy Package Code (CASTEP) [14], which is based on the first principles of the density-functional theory (DFT). The interactions between ionic and valance electron were described by ultrasoft pseudopotential method [15]. The local density approximation (LDA) of the Ceperley–Alder–Perdew–Zunger (CAPZ) functional and generalized gradient approximation (GGA) of the Perdew–Burke–Ernzerhof (PBE) were chosen for determining the electron exchange and correlation energies [16]. The plane wave energy cutoff was set as 400 eV, and the Monkhorst–Pack k-point grid of $9 \times 9 \times 1$ was selected to ensure the accuracy. The Broyden–Fletcher–Goldfarb–Shannon (BFGS) [17] algorithm was used to achieve the ground state and surface relaxation during the geometric optimisation. The convergence criteria were as follows: the energy change converged to less than $2 \times 10^{-5}$ eV/atom, maximum stress lower than 0.1 GPa, maximum force less than 0.05 eV/Å and maximum displacements within 0.002 Å.

## 3. Results and Discussion

### 3.1. Experiment

The SEM image and the EDS elemental maps of TiC combined with 316L stainless steel composites are shown in Figure 1a–d. The elemental maps of C, Ti and Fe at the same magnification clearly show both the 316L stainless steel phase and the TiC phase, which indicates that TiC-reinforced 316L stainless steel composite was successfully prepared by SLM. Figure 1e–h shows the EDS spot analysis of these particles and the diffusion area in the material. The content of Ti in Spot 2 and Spot 3 reached 77.87% and 76.29%, while that in Spot 1 at the boundary was 46.16%, which was significantly higher than that in Spot 4 (2.08%). According to the content of the Ti element, the TiC particles, 316L stainless steel matrix and the interface can be clearly distinguished. However, this is different from exploring the interfacial bonding strength of TiC reinforced 316L stainless steel composite through experiments. Therefore, we investigated the interfacial stability of TiC and γ-Fe using first principles as follows.

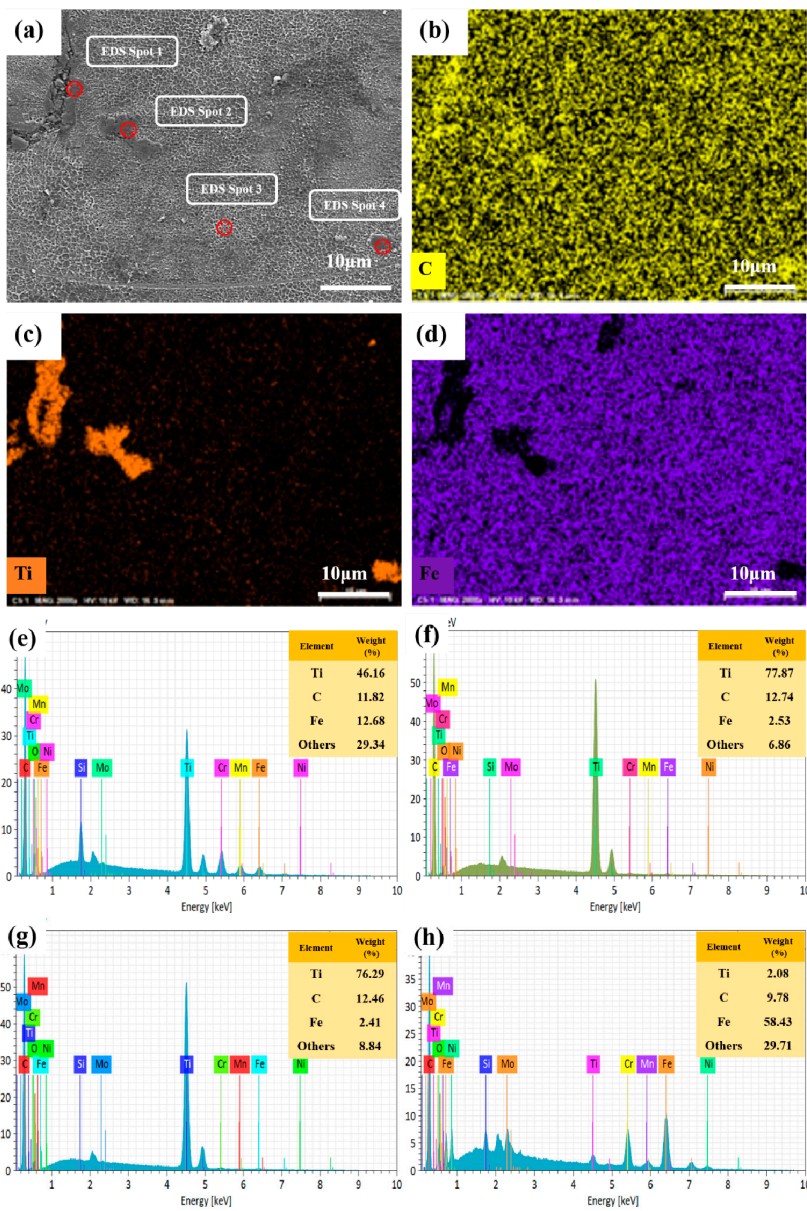

**Figure 1.** (**a**) SEM images of the TiC/316L stainless steel composites formed by SLM. EDS mapping of the (**b**) C element in (**a**); (**c**) Ti element in (**a**); and (**d**) Fe element in (**a**). Energy-dispersive X-ray spectral (EDS) spot analysis of (**e**) Spot 1; (**f**) Spot 2; (**g**) Spot 3 and (**h**) Spot 4.

### 3.2. Calculation and Simulation

### 3.2.1. Bulk and Surface Properties

Bulk Properties of TiC and $\gamma$-Fe

To guarantee the accuracy of the calculation, the lattice constant, volume, bulk modulus and formation enthalpy for TiC and $\gamma$-Fe were calculated by two different functions (LDA-CAPZ, GGA-PBE). The formation enthalpy can indicate the stability of structure. It was calculated as follows [18]:

$$\Delta H(TiC) = E_{TiC}^{bulk} - E_{Ti}^{bulk} - E_{C}^{bulk} \tag{1}$$

where $\Delta H(TiC)$ is the formation enthalpy of the TiC bulk phase, $E_{TiC}^{bulk}$ is the optimized energy of the structural unit of TiC, $E_{Ti}^{bulk}$ and $E_{C}^{bulk}$ are the energies of the individual atoms after elemental Ti and C atoms completely relaxed, respectively.

The calculated results of lattice constant, volume, bulk modulus and formation enthalpy are listed in Table 1, and are compared with the other studies and experimental results. TiC is a NaCl-type structure with a FM–3M space group. The crystal structure of bulk $\gamma$-Fe is face centred cubic with a space group symmetry of FM–3M, the lattice constants of bulk $\gamma$-Fe: a = 0.345 nm [19] are obtained by using GGA method. These lattice constants from GGA calculations are 0.1% smaller, and the results from the LDA calculations are about 1.5% smaller than other calculated and experimental values, respectively. Similarly, The GGA-determined bulk properties of TiC are also closer to the experimental results than the LDA values. Therefore, GGA-PBE is used as the exchange correlation functional for subsequent calculations.

**Table 1.** Calculated lattice constants (a), volume ($V_0$), bulk modulus (B), and formation enthalpy ($\Delta H$) of bulk $\gamma$-Fe and TiC.

| Phases | Method | A (nm) | $V_0$ (nm³) | B (GPa) | $\Delta H$ (eV/atom) |
|--------|--------|--------|-------------|---------|----------------------|
| $\gamma$-Fe | GGA$_{this\ work}$ | 0.3445 | 4.0885 | 306 | / |
| | LDA$_{this\ work}$ | 0.3395 | 4.0636 | 301 | |
| | GGA [9] | 0.3448 | 4.1010 | 314.7 | / |
| | Exp [19] | 0.3450 | 4.1060 | / | / |
| TiC | GGA$_{this\ work}$ | 0.4328 | 8.107 | 248 | −0.82 |
| | LDA$_{this\ work}$ | 0.4258 | 7.719 | 264 | −0.88 |
| | GGA [20] | 0.4320 | 8.128 | 249 | −0.76 |
| | GGA [21] | 0.4343 | 8.192 | / | / |

Surface Energy

The TiC and $\gamma$-Fe slabs should be thick enough to meet the bulk-like character interiors. With the atomic slabs increasing, more accurate results could be obtained, but a longer time and more resources are needed. Therefore, the convergence test on the slabs is performed to construct the interface calculations. The $\gamma$-Fe (001) and TiC (001) surface relaxations as a functions of termination and slab thickness are listed in Table 2, where $\Delta_{i-j}$ is the percentage of increase or decrease in the layer spacing relative to the layer spacing of the bulk material and N is the number of layers. From Table 2, it can be seen that as the number of atomic layers increases, the variation of the atomic layer spacing gradually decreases, when N ≥ 5, the surface structures with TiC and $\gamma$-Fe all show bulk-like interiors, respectively. To ensure the accuracy of the above analysis, the surface energy is calculated as follows. The surface energy is an important quantity to verify the surficial stability [22]. It can be expressed by Equation (2) [23,24]:

$$\sigma = \frac{E_{Slab}^{N} - N\Delta E}{2A} \tag{2}$$

where $E_{Slab}^N$ is the total energy of an N-layer slab, $\Delta E$ is the incremental energy obtained by $(E_{Slab}^N - E_{Slab}^{N-2})/2$, A is the surface area, N is the number of atom layers in the surface slab. In the supercell configuration for surfaces, a vacuum layer of 1 nm was selected for each surface to eliminate the interactions between the surface atoms.

**Table 2.** γ-Fe (001) and TiC (001) surface relaxations as a function of termination and slab thickness.

| Surface | Termination | Interlayer | Slab Thickness (N) | | | |
|---|---|---|---|---|---|---|
| | | | 3 | 5 | 7 | 9 |
| γ-Fe (001) | Fe | $\Delta_{1-2}$ | −4.73 | −2.06 | −2.21 | 0.39 |
| | | $\Delta_{2-3}$ | | −0.93 | −0.62 | −0.7 |
| | | $\Delta_{3-4}$ | | | 0.23 | 1.63 |
| | | $\Delta_{4-5}$ | | | | 0.58 |
| TiC (001) | C centre | $\Delta_{1-2}$ | −4.19 | −4.82 | −5.50 | −5.55 |
| | | $\Delta_{2-3}$ | | −0.90 | −0.14 | −0.81 |
| | | $\Delta_{3-4}$ | | | −1.98 | −2.21 |
| | | $\Delta_{4-5}$ | | | | −1.67 |
| TiC (001) | Ti centre | $\Delta_{1-2}$ | 0.81 | 1.04 | 1.62 | 1.08 |
| | | $\Delta_{2-3}$ | | −1.40 | −2.89 | −3.25 |
| | | $\Delta_{3-4}$ | | | −0.99 | −0.99 |
| | | $\Delta_{4-5}$ | | | | −1.98 |

The surface energy of TiC (001) and γ-Fe (001) surfaces varying with the number of atomic layers are shown in Table 3. When the thickness reached five layers, the surface energy of γ-Fe (001) converged to 2.12 J/m². The surface energy of the C centre site and the Ti centre site of the TiC surface converged to 1.72 J/m² and 1.68 J/m² with five layers. Therefore, the five-layer TiC (001) and γ-Fe (001) slabs are used in the following interface geometries, which is consistent with above results. According to the smaller surface, the more stable the interface structure, compared with C centre-sited, the Ti centre-sited is more stable.

**Table 3.** Convergence of the surface energy with respect to slab thickness.

| Layer (N) | Surface Energy (J/m²) | | |
|---|---|---|---|
| | γ-Fe (001) | TiC (001) | |
| | | C Centre Site | Ti Centre Site |
| 3 | 2.353 | 2.20 | 2.19 |
| 5 | 3.048 | 1.72 | 1.68 |
| 7 | 3.046 | 1.71 | 1.68 |
| 9 | 3.046 | 1.71 | 1.67 |

### 3.2.2. Properties of the TiC/γ-Fe Interface

TiC (001) and γ-Fe (001) Interface

The interface was modelled with a supercell slab by stacking 5-layer γ-Fe (001) on the 5-layer TiC (001), a 1 nm vacuum layer was built to avoid the interaction between the upper and lower free surface (Figure 2a). Four possible TiC (001)/γ-Fe (001) interface models with different stacking sequences (on-site and bridge-site) and different atomic arrangement sequences (C centre and Ti centre) were constructed, as shown in Figure 2b. In the top site configuration, the interfacial γ-Fe atoms are placed directly on the Ti and C atoms in the first layer of TiC side, and the C atom is located at the centre (Figure 2b(a)); the interfacial γ-Fe atoms are placed directly on the Ti and C atoms in the first layer of TiC side, and the TiC atom is located at the centre (Figure 2b(b)). In the bridge site configuration, the interfacial γ-Fe atoms are placed between the Ti and C atoms, and the C atom is located at the centre (Figure 2b(c)); the interfacial γ-Fe atoms are placed between the Ti and C atoms, and the Ti atom

is located at the centre (Figure 2b(d)), which are named as Fe-on-C centre, Fe-on-Ti centre, Fe-on-Ti centre and Fe-bridge-Ti centre.

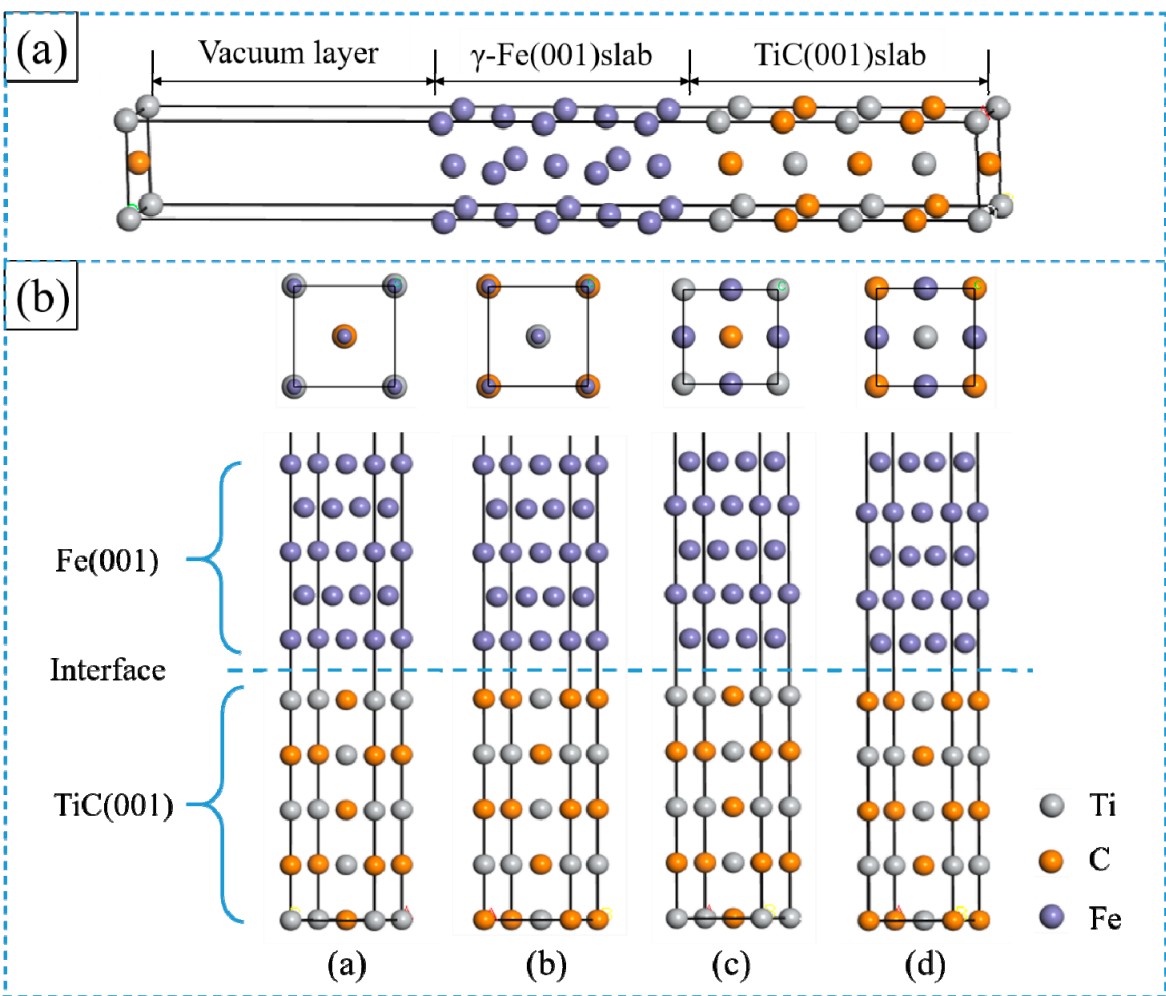

**Figure 2.** (**a**) Schematic structure for TiC/γ-Fe interface; (**b**) supercell model of TiC (001)/γ-Fe (001) interface with four different configurations: (**a**) Fe-on-C centre, (**b**) Fe-on-Ti centre, (**c**) Fe-bridge-C centre, and (**d**) Fe-bridge-Ti centre, where the blue, grey and orange spheres represent Fe, Ti and C atoms, respectively. (For interpretation of the references to colour in this figure legend, the reader is referred to the web version of this article.)

Adhesion Work

Adhesion work ($W_{ad}$) is defined as the reversible work to separate a united area interface into two free surfaces [25], which can describe the binding strength of interface atoms. The $W_{ad}$ was calculated as follows:

$$W_{ad} = \frac{1}{A}(E_{total}^{Fe} + E_{total}^{TiC} - E_{total}^{Fe/TiC}) \tag{3}$$

where $E_{total}^{Fe/TiC}$ is the total energy of fully relaxed interface, $E_{total}^{Fe}$ and $E_{total}^{TiC}$ are the total energies of relaxed isolated Fe (001) and TiC (001) slabs in the same supercell when the other slabs are replaced by vacuum. A is the surface area.

The adhesion work was calculated by using two methods. For the first one, according to the Universal Binding Energy Relation (UBER) [26], the total energies of the unrelaxed interface of TiC/γ-Fe with different interfacial distances $d_0$ were calculated. Figure 3 presents the relationship between unrelaxed $W_{ad}$ and the interfacial distances. The curve's peak represents the optimal interfacial distance and $W_{ad}$. For the second method, the $W_{ad}$ and the interfacial distance were calculated on the basis of

the fully relaxed interface (shown as Table 4). A comparison revealed that the interface distance before and after relaxation was basically consistent, but $W_{ad}$ after optimisation was significantly increased. The interface distance and $W_{ad}$ are closely related to the distribution of atoms and stacking sequence. It can be seen that under the same atomic arrangement, the interface distance of the top site is smaller than that of the bridge site, and the $W_{ad}$ is larger; under the same stacking sequence, the interface distance of Ti centre site is smaller than that of C centre site, and the $W_{ad}$ is larger. Therefore, compared with the other three interface models, the Fe-on-Ti centre had the shortest interfacial distance (0.183 nm) and the largest $W_{ad}$ (3.87 J/m$^2$), which made it the most stable interface among the four models.

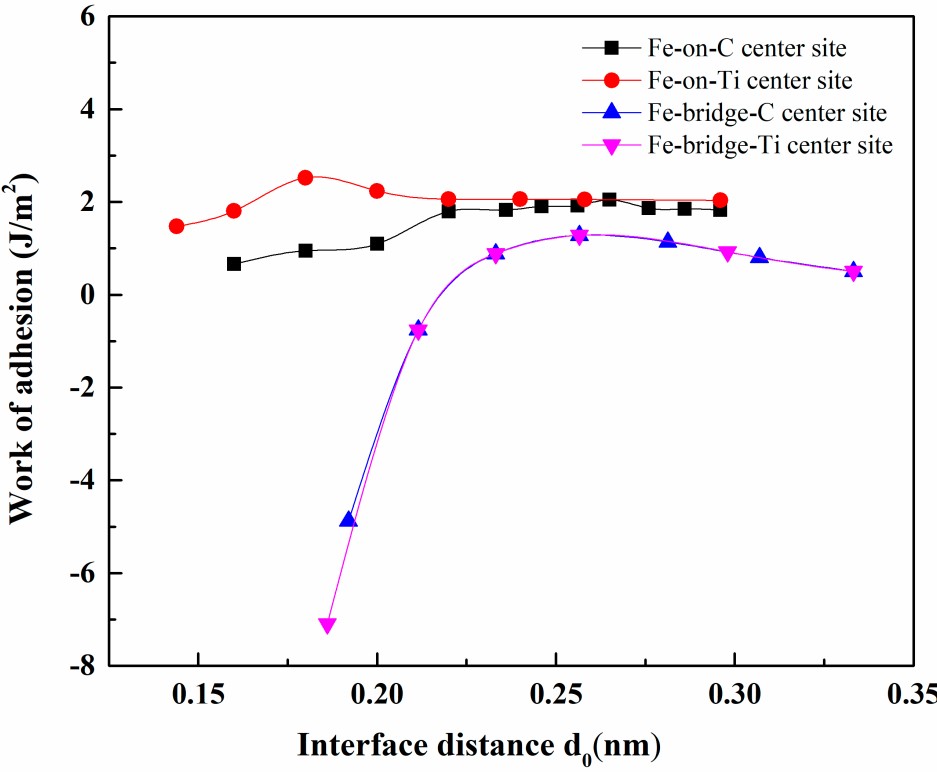

**Figure 3.** Universal binding energy curves for TiC/γ-Fe interfaces.

**Table 4.** Interfacial distance, adhesion work and interfacial energy of TiC/Fe after full relaxation.

| Termination | Stacking Sequences | After Full Relaxation | | |
|---|---|---|---|---|
| | | $d_0$ (nm) | $W_{ad}$ (J/m$^2$) | $\gamma_{int}$ (J/m$^2$) |
| C centre | on | 0.264 | 3.65 | 0.26 |
| | bridge | 0.266 | 3.03 | 0.89 |
| Ti centre | on | 0.183 | 3.87 | 0.04 |
| | bridge | 0.263 | 2.93 | 0.94 |

Interface Stability

The interfacial energy of a system can be used to describe the stability of the interface. The smaller the interface energy, the more stable the interface. The interface energy ($\gamma_{int}$) [12] was calculated as follows:

$$\gamma_{int} = \frac{1}{A}[E_{total} - N_C\mu_{TiC}^{bulk} - N_{Fe}\mu_{Fe}^{bulk}] - \sigma_{TiC} - \sigma_{Fe} \tag{4}$$

where $E_{total}$ is the total energy of interface; and $\mu_{TiC}^{bulk}$ and $\mu_{Fe}^{bulk}$ are the chemical potential of the bulk TiC and Fe atoms, respectively. $N_C$ and $N_{Fe}$ are the number of C and Fe atoms in the interface, respectively. $\sigma_{TiC}$ and $\sigma_{Fe}$ are the surface energies of the TiC and Fe surface structures, respectively. The TiC (001)/γ-Fe (001) interface energy is shown in Table 4. The interfacial energy of the Fe-on-Ti

centre model is 0.04 J/m$^2$, which is lower than that of the other models by 0.26 J/m$^2$, 0.89 J/m$^2$ and 0.94 J/m$^2$, respectively; thus, it was the most stable. Moreover, in the same arrangement of atoms, the interface energy of the top site is smaller, and its structure is more stable; in the same stacking sequence, the interface energy of Ti centre site model is smaller, and its structure is the most stable, which is consistent with previous research results.

Electronic Structure and Bonding

The electronic structure and bonding of TiC (001)/γ-Fe (001) interface atoms play a decisive role in the bond strength [20]. In addition, the stronger the bond strength is, the more stable the interfacial bonding is.

Figure 4 shows the charge density distributions of four models. It can be seen that atom redistribution, charge transfer and charge aggregation occurred at the interface. The chemical bonds are formed between Fe atoms and Ti and C atoms. The high electronic density between interfacial Fe and C atoms shows the formation of the strong interfacial covalent Fe–C bonds from Figure 4a to Figure 4d. In particular, the strong interfacial bonding is more obvious in the Fe-on-Ti centre, which indicates that Fe-on-Ti centre interfacial model is more stable. Therefore, the Fe-on-Ti centre interface has the highest W$_{ad}$ (3.87 J/m$^2$) and lowest interfacial distance (0.183 nm) in all interfacial models, which again proves that the Fe-on-Ti centre interfacial structure is the most stable.

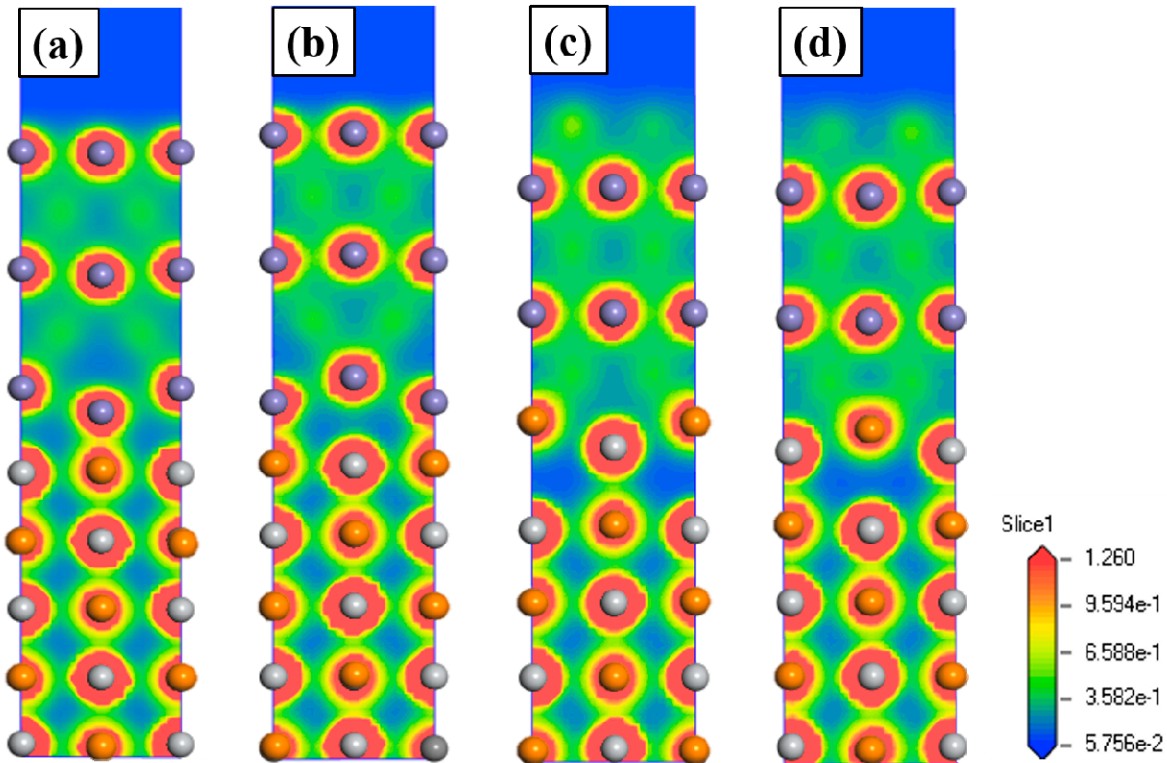

**Figure 4.** Charge density distributions for the TiC (001)/γ-Fe (001) interfaces: (**a**) Fe-on-C centre; (**b**) Fe-on-Ti centre; (**c**) Fe-bridge-C centre, and (**d**) Fe-bridge-Ti centre.

The charge density differences of four interfaces are shown in Figure 5. The red and blue regions represent the charge depletion and the charge accumulation regions, respectively. The charge density difference at the interface shows the elliptical profile and the charge distribution has a clear directionality. Figure 5a,b shows that there was a large charge accumulation around the C atoms near the interface and there was an obvious charge depletion around the Ti and Fe atoms at the interface. This phenomenon implies that a part of the charges transferred from the Fe and Ti atoms to the C atoms. In Figure 5a, Fe and Ti atoms are mostly combined with each other to form the Fe-Ti metal bond at the interface,

and more Fe-C covalent bond characteristics are obvious in Figure 5b. Fe-C covalent bonds are stronger than the Fe-Ti metal bond. Therefore, the Fe-on-Ti centre interface was more stable than the Fe-on-C centre interface. As shown in Figure 5c,d, there were no obvious charge transfers and regionalisation features at the interface.

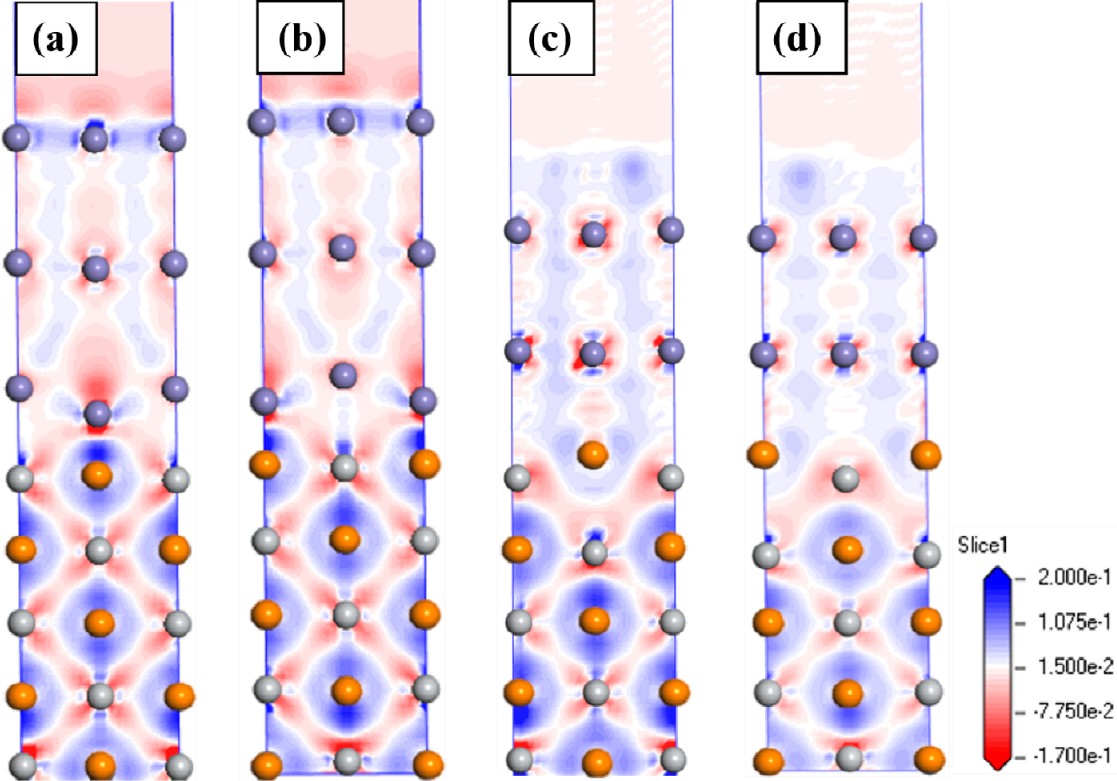

**Figure 5.** Charge density difference for the TiC (001)/γ-Fe (001) interfaces: (**a**) Fe-on-C centre; (**b**) Fe-on-Ti centre; (**c**) Fe-bridge-C centre, and (**d**) Fe-bridge-Ti centre.

In Figure 6, the partial density of states (PDOS) is shown for four interfaces. It is clear that the PDOS shapes of the first layer interfacial atoms are different from those of the inner layer atoms. It also confirms that the charge distribution is localized. The orbital curves of the interfacial Fe atoms had no obvious regionalisation, and the Fermi level of the Fe atoms was high, indicating that the interfacial Fe atoms had strong metal bonds. From Figure 6a,b, the d orbital of the first Fe layer overlapped partially with the d orbital of the first Ti layer, which showed that orbital hybridisation occurred between two atomic orbitals, and an Fe–Ti metallic bond was generated. The p orbital of the C atoms at the interface had obvious peaks near −6 eV, −4 eV and −3eV, the DOS value of the C-p atom at the interface is significant increased, and the d orbital of Fe in the first Fe layer also generated a new peak near −6 eV, indicating that a resonance peak was produced under the interaction between the Fe atoms and the C atoms, resulting in the hybridisation of the orbitals of the Fe and C atoms, thus, the Fe and C atoms near the interface formed strong covalent bonds. Therefore, it could be concluded that strong covalent bonding and weak metallic bonding generated in the interface. As shown in Figure 6c,e, the charge transfers from Fe-d orbit and Ti-d orbit to C-p orbit are relatively few than Fe-on-C centre and Fe-on-Ti centre interface. The DOS value of interfacial Fe atoms was lower in the Fermi level. There was almost no difference in the DOS of the same stacking sequences. It was demonstrated that the on-sited interfaces have more stronger stability among the four interface models, which is consistent with the previous calculations.

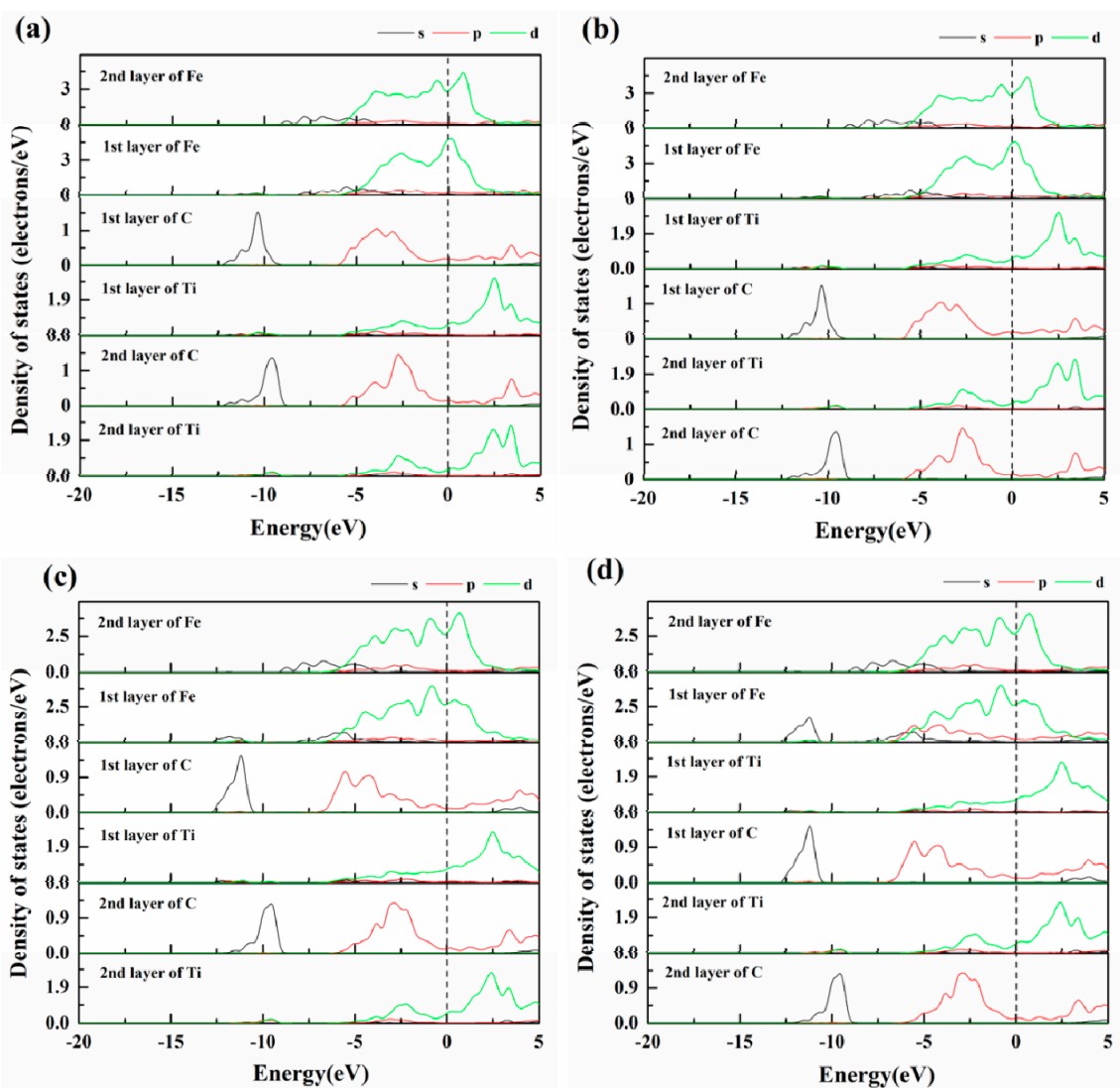

**Figure 6.** Partial density of states (PDOS): (**a**) Fe-on-C centre; (**b**) Fe-on-Ti centre; (**c**) Fe-bridge-C centre, and (**d**) Fe-bridge-Ti centre.

## 4. Analysis on TiC as Heterogeneous Nucleation of $\gamma$-Fe

The heterogeneous nucleation of matrix on carbide particles depends on the interfacial energy. The smaller the interfacial energy is, the more conducive the ferrite is to heterogeneous nucleation [27]. The interfacial energy between efficient heterogeneous nucleation TiC particles and the $\gamma$-Fe have to be lower than that of solid–liquid interface between the $\gamma$-Fe and Fe melt, about 0.24 J/m$^2$ [28]. In our calculating results, Fe-on-C centre interface model is the only one whose interfacial energy (0.04 J/m$^2$) is less than 0.24 J/m$^2$. The existence of the Fe-on-C centre interface is more beneficial for decreasing the interfacial energy, which enhances the TiC nucleation potential. AlMangour et al. [29] experimentally found that TiC has the fine grains to promote heterogeneous nucleation on the 316L stainless steel. In this paper, we provided a theoretical basis for the ferrite heterogeneous nucleation on TiC.

## 5. Conclusions

In this paper, we prepared the TiC/316L stainless steel composites by selective laser melting and analysed the microstructure of composites. The surface energy, adhesion work, interface energy, electronic structure and bonding properties of the TiC (001)/$\gamma$-Fe (001) were investigated by first principles. Four different models were compared to analyse the bonding strength and interfacial stability. The main results of this paper are as follows:

(1) The on-site interfaces have larger adhesion work and smaller interfacial energy compared with bridge-sited interfaces. The Ti centre interfaces also have larger adhesion work and smaller interfacial energy compared with C centre interfaces. Thus, the Fe-on-Ti centre interface is more stable with largest adhesion work (3.87 J/m$^2$) and smallest interfacial energy (0.04 J/m$^2$).

(2) The interfacial energy of the Fe-on-Ti centre interface of TiC (001)/$\gamma$-Fe (001) is smaller than that of solid–liquid interface between the $\gamma$-Fe/Fe. The TiC particles can act as heterogeneous nucleation substrates for $\gamma$-Fe grains from crystallography.

(3) The chemical bonding of Fe-on-C centre interface have metal characteristics. The interfacial bonding of Fe-on-Ti centre is mainly obvious Fe–C covalent bonding and shows the strongest adhesion strength.

**Author Contributions:** Methodology, formal analysis, writing—original draft, Q.W.; Conceptualisation, writing—review & editing, Z.Z., P.B.; Methodology, writing—review & editing, W.D., M.L., H.L., Y.L., B.H.; Methodology, formal analysis. L.Z., J.L. All authors have read and agreed to the published version of the manuscript.

**Funding:** This research was funded by the National Natural Science Foundation of China (Grant No. 51775521 and U1810112), the China Postdoctoral Science Foundation (2019M661068), the Key Research and Development Project of Shanxi Province (201903D121009), the Natural Science Foundation of Shanxi Province: 201801D221154, the Major Science and Technology Projects of Shanxi Province, China (No. 20181101009, 20181102012), Shanxi Foundation Research Projects for Application (201801D221234), Research Project Supported by Shanxi Scholarship Council of China (2019072).

**Conflicts of Interest:** The authors declare no conflict of interest.

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
