# Peer review of "Interfacial Stability of TiC/γ-Fe in TiC/316L Stainless Steel Composites Prepared by Selective Laser Melting: First Principles and Experiment"

_metals, doi:10.3390/met10091225_

Round 1
Reviewer 1 Report
A good and consistent work. The presentation is also good. However, some details could be improved. Please see the list below.
- Line 61: “In addition, many studies have discussed the heterogeneous nucleation potential of Fe on TiC atoms and grain refinement by experiment, which is relatively scarce for theoretical analysis.” Which studies? References are missing.
- What for was the composite prepared by SLM used in this research? For obtaining data for the calculations? For validation of calculated results? Please provide an explanation.
- Lines 68-70: Parameters are incomplete. Laser beam diameter (focal diameter) and hatch spacing or track overlapping are missing.
- Lines 90-91: “Therefore, the distribution of TiC on 316L stainless steel was uniform.” Figure 1 does proof that the distribution of TiC in the matrix was homogeneous. The area shown in the figure is too small for that. However, the goal of this work was not production of a homogeneous composite. Therefore, the statement about homogeneity could be omitted.
- Lines 99-101: “In order to guarantee the accuracy of the calculation and the rationality of the models, the lattice constant, volume, bulk modulus and formation enthalpy for TiC and γ-Fe, which were calculated by two different functions (LDA-CAPZ, GGA-PBE).” Something is wrong with the structure of this sentence.
- Line 101: the abbreviations LDA and CAPZ were not explained.
- Angstrom: Use of non-SI units is inappropriate. Please use nanometers m instead of Angstrom.
- Line 134: “10Å” Between the value and the unit there shell be a free space. Please, check in the entire manuscript. The same applies to the symbol of a quantity and the unit.
- Figure 2: a) The two shades of gray used for Ti and C atoms look quite similar. A more distinct difference between colors would be better. b) Please mark the Ti and C atoms in the figure.
- Line 214: Wad – “ad” should probably be written as subscript.
Author Response
Thanks for your comments,please see the attachment.

Reviewer 2 Report
The paper is well written and interesting. It deals with TiC layers on top of Fe,
where both systems crystallize in Fm-3m space group. It made it easy to simulate using general ab-initio code. As I understand CASTEP is pseudopotential code, so the information about pseudopotentials should be supplied. From their manual on castep site it can generate pseudopotentials
on the fly, but the conditions should still be mentioned in the paper.
I assume that the authors just used the automatic approach, which in case
of simple 3d metals and C might be accurate enough. Summarizing, I think the paper can be published after authors add the details about calculations conditions.
Author Response

(The authors gave the same response as above.)

Reviewer 3 Report
The paper deals with the fabrication of metal matrix composites based on 316L using SLM, and is focused on first principle calculations. The paper is well organized and deserves publication. I have some minor comments for the authors:
-Please add the name and company of any equipment you used
-In figure 1, what is the unindexed peak between Si and Mo?
-Please correct small typos in the paper as in lines 66 and 83-84 (verbs).
Author Response

(The authors gave the same response as above.)

Reviewer 4 Report
The current study provides a modeling (first principle) based approach to understand the heterogeneous nucleation potential of gamma Fe on TiC particle of an SLM produced 316L/TiC composite. Their approach consists of the 4 different interfacial models for the stacking sequence of TiC/gamma-Fe. Although the authors have performed a rigorous modeling calculation, the experimental part lacks strength. The following are the specific comments which should be addressed before the publication of the manuscript.
- Line 69- Experimental section- what kind of atmosphere was used during the SLM process.
- Line 71- Provide the Operating condition (Voltage current etc) of the SEM operation.
- Line 76- Before using the abbreviation, please define it. (BFGS)
- Line 82- What is the basis of the claim that TiC is uniformly distributed when large chunks can be seen in EDS map of Ti as well as in spot analysis.
- Authors also should provide about the phases in the final microstructure after SLM process to indicate the phases (esp Fe, can be done via XRD or EBSD.)
- Line 107 -Please provide the symbols before using them in Table 1.
- Line 109- Please use the proper notation for Fm-3m with “bar” rather than “negative symbol”.
- The color coding used in Figure 2 for Ti and C is both grey and not distinguishable.
- Conclusion line 3 is not making sense. Please correct it.
